# Inferring Time-Lagged Causality Using the Derivative of Single-Cell Expression

**DOI:** 10.3390/ijms23063348

**Published:** 2022-03-20

**Authors:** Huanhuan Wei, Hui Lu, Hongyu Zhao

**Affiliations:** 1SJTU-Yale Joint Center for Biostatistics, Shanghai Jiao Tong University, 800 Dong Chuan Road, Shanghai 200240, China; weihh123@sjtu.edu.cn; 2Department of Biostatistics, Yale University, New Haven, CT 06520, USA

**Keywords:** causal inference, time-lagged regulation, gene regulatory network, single-cell data

## Abstract

Many computational methods have been developed to infer causality among genes using cross-sectional gene expression data, such as single-cell RNA sequencing (scRNA-seq) data. However, due to the limitations of scRNA-seq technologies, time-lagged causal relationships may be missed by existing methods. In this work, we propose a method, called causal inference with time-lagged information (CITL), to infer time-lagged causal relationships from scRNA-seq data by assessing the conditional independence between the changing and current expression levels of genes. CITL estimates the changing expression levels of genes by “RNA velocity”. We demonstrate the accuracy and stability of CITL for inferring time-lagged causality on simulation data against other leading approaches. We have applied CITL to real scRNA data and inferred 878 pairs of time-lagged causal relationships. Furthermore, we showed that the number of regulatory relationships identified by CITL was significantly more than that expected by chance. We provide an R package and a command-line tool of CITL for different usage scenarios.

## 1. Introduction

Single-cell RNA sequencing (scRNA-seq) is a technology capable of measuring the expression level of RNA at the single-cell resolution [1]. Rapidly growing scRNA-seq data open the door to a sufficiently powered inference of causality among genes. Several computational methods have been developed for causal inference from cross-sectional data (e.g., [2,3,4]) or time-series data (e.g., [5]). These methods have been applied with some success to biological data [6,7,8].

With reference to the time factor in causal inference, causal relationships among genes can be categorized into instant relationships and time-lagged relationships. In this study, we focus on the second. A time-lagged relationship is illustrated in Figure 1. The expression level of gene *i* at $t_{0}$ will affect the expression level of gene *j* at $t_{1}$, which is denoted by the black arrow connecting gene *i* with gene *j*. Note that with a time-lagged relationship, the expression level of gene *i* may not be related to the expression level of its target gene *j* at a specific time $t_{0}$.

There are two main challenges to inferring time-lagged causality for scRNA-seq data: the collection of longitudinal data and the presence of latent variables. First, it is difficult to continuously monitor the whole transcriptome within one cell. Of note, even when cells can be sequenced at different time points [9], such data cannot be considered as real time-series data because they capture different cells instead of the same set of cells. In Figure 1, the connections between time points are broken because distinct cell populations are studied. That is, we are not able to trace the evolutions of cells across different time points. We refer to such data as multi-trace data, where cells are collected from different time points. We will investigate whether such data may help us infer causality among genes through simulation studies. The reason why continuous measures matter is that there are natural confounders when inferring time-lagged causality for cross-sectional scRNA-seq data. For every cell, only the expression levels of genes (the colored circles in the bottom part of Figure 1) at time point $t_{1}$ can be obtained from sequencing. For time-lagged relationships, the expression levels of the causal genes at the previous time point, i.e., $t_{0}$, act as confounders between the current expression levels of the causal genes and their targets’ expression levels. As shown in Figure 1, the time-lagged causal gene pairs are not linked directly. If the expression levels of causal genes at previous time points are not considered, the association between the current expression levels of the causal genes and their targets can be low or even in the opposite direction. Throughout this paper, we refer to such confounders as natural confounders (red rectangle in Figure 1). This problem was noted previously [10] but has not been well addressed in the existing literature.

The second challenge is that unmeasured variables, also referred to as latent variables, are common in scRNA-seq experiments. scRNA-seq can capture the expression levels from 2000 to 6000 genes in a cell, where many genes with low-expression levels may not be captured. Additionally, the causal path from one gene to another often involves many biological molecules which cannot be detected by scRNA-seq, such as proteins, metal ions, and saccharides. Together with low-expression genes, these latent variables are common for scRNA-seq data. However, many existing methods for causal inference assume the absence of latent variables, and as a result, may have difficulty in inferring causality from scRNA-seq data.

Here, we propose CITL (causal inference with time-lagged information), a method to infer the time-lagged causal relationships among genes in scRNA-seq data capable of overcoming the two challenges mentioned above. CITL uses RNA velocity information inferred from scRNA-seq data to estimate the changing expression levels of genes. By assessing the conditional independence between the changing and current expression levels, CITL reduces the interference by natural confounders. CITL can more accurately infer time-lagged relationships than a commonly used cross-sectional causal inference algorithm, the PC-stable algorithm [4] in simulations. Compared with [8], which also uses RNA velocity to infer causality, CITL is more stable in simulation studies and may better identify time-lagged causality from extensive real data. On real scRNA-seq data, we show the concordance between the time-lagged causal relationships inferred by CITL and regulatory pathways curated from the literature. Our results suggest that time-lagged causality may represent the relationships involving multi-modal variables.

## 2. Results

### 2.1. The Time-Lagged Assumption Helps Overcome the First Challenge

CITL focuses on the causal relationship without natural confounders. As shown in Figure 1, the natural confounder (the red rectangle) disrupts the relationships between the current gene expression levels (red dotted line). However, the green bar, representing the relationship between the current and changing gene expression levels, is not influenced by the red rectangle. The changing expression level of gene *j* (the triangle in Figure 1), unlike the current or subsequent gene expression levels, inherits information from the red rectangle only through its cause, the current expression level of gene *i*. In this situation, the red rectangle is no longer a natural confounder for the green bar. That means the “strong correlation” described in the Time-lagged assumption will not be influenced by natural confounders. Given this property, CITL assesses the dependency between the current and changing gene expression levels to reduce the influence of natural confounders. The following simulation results prove this. The elucidation of the simulation set-ups and the approaches for comparison was in Section 4.

#### 2.1.1. Simulation Results of Approach 0

The performance of Approach 0 largely depends on the threshold of Pearson’s correlation coefficient. We tested its performance at 18 thresholds from 0.1 to 0.9 through 500 simulations for each setting. Figure 2 summarizes the performance of Approach 0 in single-trace simulations (top row in Figure 2) and multi-trace simulations (bottom row in Figure 2). In single-trace simulations, the precision increased, and the recall decreased, as the threshold increased for both finding edges and determining causal directions. The more stringent the threshold was, the more accurate Approach 0 was, but the fewer edges Approach 0 could find. When the threshold was around 0.2, Approach 0 achieved the highest F-measure in single-trace simulations. In contrast, the highest F-measure of Approach 0 in multi-trace simulations was achieved when the threshold was around 0.75. The overall performance of Approach 0 in multi-trace simulations was much worse than that in single-trace simulations. This suggests that multiple traces cause many false positives for both finding edges and determining causal directions in Approach 0.

#### 2.1.2. Comparisons CITL with PC-Stable and Its Variant Approaches

The simulation results for the single-trace scenario for the other approaches are summarized in Table 1. For finding edges, Approach 1 achieved the lowest precision, which is expected as PC-stable applied to current expression levels will miss time-lagged causal edges via single-trace data. The recall of Approach 2 was lower than others, which suggests that the natural confounders in Approach 2 clearly influenced the discovery of casual edges. Approach 3 and CITL derived the same UG, which performed the best in both recall and F-measure, demonstrating that changing information is useful when identifying edges between causal pairs from single-trace data. When determining the causal direction, CITL performed the best, and Approach 1 had the worst performance. Both Approach 1 and Approach 2 performed worse than Approach 3 in recall and F measure, indicating that natural confounders influence the determination of causal directions. CITL was better than Approach 3 for all three metrics, demonstrating that CITL was the most effective in determining causal directions than the assumptions of PC-stable. As for multi-trace simulations, we obtained similar results, as shown in Appendix A. Additionally, CITL was more stable than other approaches in simulations for different combinations of $C_{variance}$ and $I_{variance}$ (Appendix A). In summary, CITL had the best performance among the approaches and was less sensitive to the type of data applied.

As for the results for inferring instant causality, the type of simulation affected the determination of directions. Compared with the results of single-trace simulations (Appendix A), the F-measure for determining the directions of Approach 1 decreased, while that of CITL increased in multi-trace simulations (Appendix A). This suggests that natural confounders, the previous expression levels of causes, could have a larger effect on instant causal relationships for multi-trace data. In this case, CITL was still a good choice for identifying instant causality.

#### 2.1.3. Comparisons with Scribe under Different Simulation Settings

We also evaluated the performance of Scribe. In our 50-node simulations, Scribe ran much slower (about 1470 s) than single-threaded CITL (less than 10 s). Consequently, we only applied Scribe to one replicate of each simulation. The edges discovered by CITL were compared with the top-RDI edges of the same number (Table 2). Besides, we evaluated Scribe with the area under the receiver operating characteristic curve (AUROC) and the area under the precision-recall curve (AUPR). Scribe needs an appropriate threshold of RDI to discover causal edges in the single-trace simulation. In addition, Scribe and Approach 0 performed similarly in both single-trace and multi-trace simulations (Appendix A). Compared with Scribe, CITL was more stable and correctly oriented more edges in the multi-trace simulation.

We further compared Scribe with CITL through simulations conducted as previously described by [8]. In the simulation, the noise in the differential equations was set to three levels as mentioned in Section 4.3; this represented the randomness of the causal effect, the temporal fluctuation, and random error. The results are shown in Appendix A. Under the low-noise and high-deviation setting, both Scribe and CITL performed well. Similarly, both approaches were affected by the high-mean noise. Considering the results in Qiu’s and our simulation set-up, CITL’s performance was more stable, and its runtime was much shorter.

#### 2.1.4. CITL with Asynchronous Regulations

The asynchronous regulations are time-lagged regulations with different extrapolation time steps. Therefore, the changing expression level of each asynchronous gene should be its reaction velocity multiplied with its corresponding extrapolation time step if the velocity is stable. However, the extrapolation time steps of all regulations are not available. CITL used RNA velocity to estimate the changing expression levels of all genes, assuming that the extrapolation time step used in RNA-velocity estimations is constant across cells and genes (same as [8]). The assumption could lead to a bias estimation of the changing expression levels of asynchronous genes. To evaluate the impact of asynchronous regulations on CITL, we designed simulations where the amount and degree of asynchronous regulations varied. As shown in Figure 3, in all 25 set-ups of asynchronous simulations, none of the metrics were significantly different from those of the control. This shows that the asynchronous regulations had less impact on the performance of CITL for both discovering time-lagged relationships and determining causal directions. Under such a bias estimation of the changing gene expression, CITL maintains its ability to infer time-lagged causal relationships.

#### 2.1.5. CITL with Latent Variables

The second challenge has less impact on causal orientation when adopting the Time-lagged assumption. As shown in Figure 1, both the red dotted line and the green bar are undirected. For the former, the relationship between the current gene expression levels can be oriented only by PC-stable’s assumptions, including *causal sufficiency assumption*, which requires that all variables are measured. However, the *causal sufficiency assumption* is violated as described in the second challenge. For the latter, the relationship between the current and changing gene expression levels can be oriented according to the Time-lagged assumption. Since the current gene expression level precedes the changing gene expression level in time, the temporal order of these two types of gene expression levels is defined as the causal direction in the Time-lagged assumption. It enables CITL to orient causal edges without causal sufficiency, which releases the disturbance derived from the second challenge.

To evaluate the impact of latent variables on CITL to infer time-lagged causality, we performed single-trace simulations by randomly removing 0%, 10%, 30%, and 50% of the total genes. As illustrated in Figure 4a,b, as the number of latent variables increased, the performance of all approaches reduced for both finding causal edges and determining causal directions. This showed that latent variables had a negative effect on all approaches as expected. CITL performed the best across all the simulation settings. We used ADD to evaluate how well an approach inferred the causal directions in the presence of latent variables. The distribution of ADD in the simulations is shown in Figure 4c. The ADD of CITL concentrated at a higher level, while other approaches were not stable. This shows that CITL is more robust than other approaches. Similar results were obtained for multi-trace simulations (Appendix A).

### 2.2. Applications to Real Data Sets

#### 2.2.1. Evaluation of the Information in RNA Velocity for Inferring Causal Relationships

For real data sets, we estimated the changing expression levels and the subsequent expression levels by RNA velocity. Before adopting the estimated $X^{cha}$ to infer causality, we investigated how much information it contained. First, we observed that using the estimated $X^{cha}$ to calculate the correlation led to different correlated pairs than when using $X^{cur}$ in data set 1 (Figure 5a) and data set 2 (Figure 5b). This suggests that the information for the estimated $X^{cha}$ was different from that of $X^{cur}$. A similar method recently developed drew the same conclusion [11]. Second, we applied Approach 0 to both data sets, and the resulting networks showed that the distributions of indegree and outdegree were very different (Appendix A). In addition, the molecular function of low-outdegree genes was associated with gene regulation (Appendix A). Taken together, the unique information of the estimated $X^{cha}$ suggests that CITL could use RNA velocity to estimate the changing expression levels.

#### 2.2.2. Causal Inference Using CITL on Real Data Sets

We applied CITL to data set 1 and data set 2 with 2508 and 878 time-lagged causal pairs (TLPs) inferred, respectively. We also applied PC-stable on the data sets with current-only expression data and compared the gene pairs inferred by PC-stable to TLPs. For computational efficiency, the value of *k* for both CITL and PC-stable was equal to the square root of the number of genes for each data set. A total of 3998 and 4459 pairs were inferred by PC-stable from data set 1 and data set 2, respectively. In data set 1, only four gene pairs were found by both approaches, and there was no overlap for data set 2. These results suggest that CITL infers different types of causality from previous methods that only used the current expression level of genes.

#### 2.2.3. CITL Accurately Infer Time-Lagged Causal Pairs

Because we do not know the ground truth for time-lagged causality, we investigated the biological relationships of TLPs to evaluate the performance of different methods. Pathway Studio (http://www.pathwaystudio.com/ (accessed on 20 October 2020)) enables searching interactions between molecules, cell processes, and diseases from the literature. Almost any pair of two genes could be related, directly or indirectly, through Pathway Studio. Each interaction is annotated by a sentence from the literature. Not all interactions are regulatory, such as binding. We reviewed the annotation of every searched interaction to find TLPs with regulatory interactions. For the regulatory interactions, we divided them into two categories. The “PROT” type refers to interactions that only involve proteins, such as increasing or reducing protein activity, co-activating or antagonizing, and phosphorylating or dephosphorylating. The “TRSC” type refers to interactions relating to proteins regulating the transcription of specific genes, including activation and repression. Considering manually filtering interactions taking considerable time, we only investigated the biological functions of a subset of the pairs.

In the following, we describe how we chose the subset of TLPs to consider. The single-trajectory developmental cells in data set 2 make it easier to visualize time-lagged relationships than multi-trajectory differentiating cells in data set 1. Therefore, we focus on the TLPs in data set 2, where 37 transcription factors were involved in 68 TLPs. Transcription factors (TFs) were taken from the TRRUST database, a repository of curated TF–target relationships of human and mouse [12]. We investigated these 68 TLPs in Pathway Studio and manually checked the interactions of each TLP.

All the 68 pairs had indirect relationships, forming paths with one or more intermediates. Most of the interactions among these paths were “non-regulatory”. We focused on the regulatory paths that ended with a TRSC interaction, since the causality among genes’ transcripts, rather than proteins, was of interest in scRNA-seq. In total, 14 TLPs with regulatory relationships (rTLPs) and their regulatory paths are shown in Table 3. The interaction types are listed from the left of the corresponding path to the right. CITL achieved an accuracy of 0.93 (13/14) for correctly inferring the causal directions of rTLPs. The regulatory effect (activation or repression) of 11 pairs was correctly described. Only one rTLP was assigned an inconsistent direction with its path (the cur_cha of *MAGED1*—*EOMES* was −0.19).

To evaluate the significance of the accuracy of CITL, we first investigate how likely it is that a random gene will be the target of a TF. We randomly chose 11 TFs from the 37 TFs and investigated their regulatory relationships with randomly selected genes. For each TF, a randomly selected gene was assigned as its effect. Then, the functional connection between the gene pair, referred to as randomly selected-and-direction-assigned pair (RAP), was searched using Pathway Studio. Like the TLPs inferred by CITL, most RAPs did not have regulatory relationship. To find a gene that had a regulatory relationship with each TF, we searched 35 RAPs. In the 11 RAPs with regulatory relationships (rRAPs), only two rRAPs’ assigned directions were consistent with their known causal directions. Therefore, we speculate that, for a TF, there are more upstream genes than downstream ones after excluding non-regulatory genes. We compared the accuracy of CITL to the accuracy of random selection using Fisher’s exact test. The *p*-value of the test was 0.00024, suggesting the excellent performance of CITL.

#### 2.2.4. A Example of Time-Lagged Causal Pair

We highlight a time-lagged causal pair, “*MLLT3* → *FLRT3*” in Figure 6. “*MLLT3* → *FLRT3*” is a gene pair with a small negative cur_cur correlation (−0.19) and a large negative cur_cha correlation (−0.99). Though the correlation between the current expression levels was weak, this gene pair showed a strong negative correlation in terms of time-lagged association. The inconsistency can be explained as follows. The decrease in *FLRT3* in stages 5 and 6 is due to the high expression level of *MLLT3* in stages 3 and 4 (Figure 6c). We further investigated whether this pair had a transcriptional causal relationship. *MLLT3* participated in the activity of E2F1 protein [13], which could repress WNT signaling [14]. WNT signaling could control the expression of *FLRT3* [15]. In short, *MLLT3* could repress the expression of *FLRT3*, which is consistent with the result of CITL.

#### 2.2.5. CITL Overcomes the Limitations of scRNA-seq

Indirect regulations involved more biological reactions than direct regulations, making it more reasonable to consider time-lagged relationships. Due to technical limitations, some intermediates in the indirect regulations were difficult to detect by scRNA-seq. Therefore, researchers often have to deal with indirect relationships. Here, the only path in which all genes were detected was “*YBX1* → NF-κ-B → *EPS8*”. The protein encoded by *YBX1* can activate NF-κ-B signaling [16], which then induces the transcription of EPS8 [17]. The cur_cur correlations between “*YBX1*–*NFKB1*”, “*NFKB1*–*EPS8*”, and “*YBX1*–*EPS8*” were −0.72, −0.70, and 0.04, respectively. None of these could explain the relationship between *YBX1* and *EPS8* in the literature. On the other hand, the cur_cha correlation between “*YBX1*–*EPS8*” was 0.95, which is consistent with the relationship between the genes. The results demonstrates that indirect relationships can be time-lagged relationships and that CITL is a better way of discovering these relationships.

Furthermore, some intermediates were not RNA at all. As shown in Table 3, most paths involved PROT steps. The best way to describe “*YBX1* → *EPS8*” would be through the expression level of *YBX1*, the protein activity of NF-κ-B and the expression level of *EPS8*. Although many single-cell multi-omics technologies have been developed, none of these can ensure that all of the necessary molecules in each cell are quantified. However, CITL accurately inferred indirect relationships without any protein-level information. Consequently, the CITL, discovering time-lagged relationships, was more practical than previous methods which focused on instant interactions in scRNA-seq data.

## 3. Discussion

The changing expression levels of genes are crucial to CITL. Thus, the approach used to estimate these levels can have major impact on the results. There are two main challenges to correctly estimate the changing expression levels with RNA velocity. First, scRNA-seq technologies have limitations on quantifying transcripts. The quality of raw data is of great importance to results. Second, there is no gold standard to evaluate the estimated changing expression levels. Recently, another method was developed to infer RNA velocity [18]. The estimation of RNA velocity depends on the method to use and its tuning parameters [18,19]. The more accurate the estimation of the changing expression levels of genes are, the more reliable CITL will be. Despite the two obstacles, RNA velocity has proved its usefulness in estimating the transcriptional changes of genes in many applications [20,21,22]. Our simulations with multiple combinations of the variances and asynchronous regulations show the robustness of CITL. Additionally, Qiu et al. investigated three approaches to derive single-cell time-series data and concluded that RNA velocity was the most appropriate way to estimate real time-series data through simulations [8].

Both Scribe and CITL used RNV velocity to infer causal gene regulatory networks. There are two reasons that CITL is more applicable for the task. First, single-cell samples in a real data set are likely from different populations or developmental stages, forming multi-trace data. By using the changing gene expression levels rather than the subsequent, CITL alleviates the influence of natural confounders. This could be why CITL is more stable than Scribe in the multi-trace simulations, which is practical in real single-cell data. Second, thousands of genes form a complicated network in a cell. CITL runs dozens of times faster than Scribe so that it is capable of searching the complete network.

A drawback of CITL is that it cannot distinguish whether the type of a relationship is time-lagged or instant. In biology, the relationships between genes can be a mixture of time-lagged and instant relationships. If we can confirm the interactional type of each gene pair and adapt CITL to the type, the overall accuracy may be greatly improved.

## 4. Materials and Methods

### 4.1. Causal Inference with Time-Lagged Information (CITL)

We make the following assumption for our causal inference:

**Assumption** **1.**
*Time-lagged assumption: if the current expression level of gene i $X_{i}^{cur}$ is strongly correlated with the changing expression level of gene j $X_{j}^{cha}$, then gene i is inferred to be the cause of gene j in a time-lagged manner.*


A strong correlation means that $X_{i}^{cur}$ and $X_{j}^{cur}$ are dependent on other variables, which can be assessed by the conditional independence (CI test). With this assumption, the $X^{cha}$ of a gene is not related to its $X^{cur}$ value but is correlated with the $X^{cur}$ of its causal genes. Therefore, the $X_{i}^{pre}$, the natural confounder between $X_{i}^{cur}$ and $X_{j}^{cur}$, does not directly influence $X_{j}^{cha}$. $X_{i}^{pre}$ can influence $X_{j}^{cha}$ only through $X_{i}^{cur}$, which means it is not a natural confounder for the correlation between $X_{i}^{cur}$ and $X_{j}^{cha}$.

RNA velocity [19] offers a way to estimate gene expression changes based on spliced mRNA and unspliced RNA information. CITL uses RNA velocity for a unit of time as the changing expression level $X^{cha}$ and the extrapolated expression levels in a unit of time as the subsequent expression level $X^{sub}$. Note that we use a fixed unit time in this manuscript as an approximation, although the length of time that different genes exert effects on other genes may differ. For consistency, we used the same parameters described in [19] to calculate RNA velocity.

To infer time-lagged causal relationships, CITL first constructs an undirected graph (UG) through both $X^{cur}$ and $X^{cha}$. Each node in the UG represents the $X^{cur}$ or $X^{cha}$ of a gene. Each edge in the UG represents the dependency between the $X^{cur}$ (or $X^{cha}$) of a gene and that of another gene. The dependency is assessed by CI test conditional on at most *k* (≤ the number of nodes *n*) genes. For the simulations and applications in this article, the *k* was set at the square root of *n*, and the type one error (α) of the CI test was set at 0.05. CITL then focuses on the edges linking the $X^{cur}$ of some genes to the $X^{cha}$ of others. If the $X^{cur}$ of a gene is linked to the $X^{cha}$ of another in the UG, the former gene is assigned as the cause of the latter gene. We note that the $X^{cur}$ (or $X^{cha}$) of some genes can be linked to each other. We assume that these connections do not represent time-lagged relationships. Thus, they are not the focus of this work. We provide an R package (*CITL*) at https://github.com/wJDKnight/CITL-Rpackage (accessed on 30 November 2021) and an open-source command-line tool of CITL at https://github.com/wJDKnight/CITL (accessed on 30 December 2021). Tutorials about data preparation and using CITL are also provided in the repositories.

### 4.2. Comparisons with Other Methods

We compared the performance of CITL versus a commonly used causal Bayesian network method, PC-stable [4], and a recently published causal inference method for scRNA-seq data [8], Scribe. PC-stable first constructs a UG as well. Therefore, CITL adopts the same strategy to construct the UG by using the *bnlearn* package [23]. The difference is that after the UG was built, PC-stable used probabilistic dependency to determine the causal direction under three assumptions: *Causal Sufficiency*, *Causal Markov Assumption*, and *Faithfulness* [2,24]. We compare the performance of CITL with the PC-stable through simulations under different approaches of analyzing scRNA-seq data, as detailed in the following. The parameters (*k* and α) of these approaches and CITL for constructing the UG are identical.

Approach 1: PC-stable, using only $X^{cur}$. This is the simple adoption of the causal inference methods to scRNA-seq data. As discussed above, it will not be able to infer time-lagged relationships. We include this approach to assess the lack of power to identify time-lagged relationships with only $X^{cur}$.Approach 2: PC-stable, using $X^{cur}$ and $X^{sub}$, where $X^{sub}$ is the extrapolated expression levels at the subsequent time point. For this approach, PC-stable is applied to time-lagged data. However, natural confounders may still exist between $X^{cur}$ and $X^{sub}$. Consequently, we consider this scenario to assess the effect of natural confounders on causal inference.Approach 3: PC-stable, using $X^{cur}$ and $X^{cha}$ but without the Time-lagged assumption. This approach infers causality by PC-stable itself based on three assumptions [4]. We include this approach to investigate the usefulness of the Time-Lagged assumption.

We note that any method which can identify a strong correlation between $X^{cur}$ and $X^{cha}$ may be suitable for the proposed framework. In addition to the above three approaches, we also consider another approach, Approach 0, which is the simplest version of the proposed framework using Pearson’s correlation coefficient to discover a strong correlation between $X^{cur}$ and $X^{cha}$. If the absolute value of Pearson’s correlation coefficient between $X_{i}^{cur}$ and $X_{j}^{cha}$ is above a threshold, we infer gene *i* as the cause of gene *j* as baseline prediction.

We also consider a recently published causal inference method for scRNA-seq data [8], Scribe. It uses restricted directed information (RDI) to evaluate the causal effect of the current expression levels on the subsequent expression levels. Similar to Approach 2, Scribe assumes that if the RDI of $X_{i}^{cur}$ and $X_{j}^{sub}$ is higher than a threshold, gene *i* is the cause of *j*. The default values of the parameters of Scribe were used in simulation studies.

### 4.3. Simulation

Some experiments sequence cells at one time point while others sequence cells at multiple time points. We refer to the former as single-trace data and the latter as multi-trace data. We considered both scenarios in our simulations. For single-trace data, we simulated 3000 cells. For multi-trace data, we simulated from three traces, with each trace having 1000 cells. We carried out 500 simulations for each set-up. For each simulation, we randomly generated a causal graph $G_{true}$ that contained 50 nodes (genes) and 50 directed edges on average. The probability of an edge between nodes was 4.1% (50 edges in total), and its direction was randomly assigned. Time-lagged relationships were simulated in the following manner:(1)$$\begin{array}{c} \begin{array}{cc} X_{i}^{cur} & =f_{1}\left(X_{i}^{pre}\right)+f_{2}\left(causalpre\left(X_{i}\right)\right)+e^{cur}\\ X_{i}^{sub} & =f_{1}\left(X_{i}^{cur}\right)+f_{2}\left(causalcur\left(X_{i}\right)\right)+e^{sub}\\ X_{i}^{cha} & =X_{i}^{sub}-X_{i}^{cur} \end{array} \end{array}$$

For each cell, we simulated four values related to each of the 50 genes’ expression levels, including previous $X_{i}^{pre}$, current $X_{i}^{cur}$, subsequent $X_{i}^{sub}$, and changing $X_{i}^{cha}$. Based on the collected values of $X_{i}^{pre}$ and the causal graph, $X_{i}^{cur}$, $X_{i}^{sub}$, and $X_{i}^{cha}$ were generated through Equation (Equation 1) using $causalpre(X_{i})$ (the previous values of the causes of $X_{i}$) and $causalcur(X_{i})$ (the current values of the causes of $X_{i}$). $e^{cur}$ and $e^{sub}$ represent standard Gaussian noise N(0,1). They are the primary sources of variation in our simulation, referred to as $C_{variance}$.

Here, we used linear functions to describe time-lagged relationships. The coefficient of $X^{pre}$ or $X^{cur}$ in the linear function $f_{1}\left(\right)$ was 0.8, simulating the transcripts of genes that spontaneously degrade over time. The coefficients of all causal genes in $f_{2}\left(\right)$ were set to 1, assuming all causal genes had the same effect on their effector genes. In addition, we assumed that $X_{i}^{pre}$ did not interact with $causalpre(X_{i})$, meaning there was no feedback, and we could add $f_{1}\left(X_{i}^{pre}\right)$ and $f_{2}\left(causalpre\left(X_{i}\right)\right)$.

For single-trace data, $X_{i}^{pre}$ was assumed to follow a log-normal distribution with a constant mean and standard deviation (ln(X)∼N(0,0.04)). We chose a log-normal distribution because RNA-sequencing data are often skewed rather than having a normal distribution. For every run of the multi-trace simulation, we simulated three separate sets of data (trace). In each trace, $X_{i}^{pre}$ followed a log-normal distribution (ln(X)∼N(μ,0.04)) with its mean (μ) randomly drawn from a uniform distribution between 0 and 2. It is the second source of variation in our simulation, referred to as $I_{variance}$. Then, the three traces were merged into one data set, taking into no account of the trace information. For simulations considering latent variables, we randomly removed a certain proportion of the nodes (genes) after data generation.

In our simulations, we also investigated whether CITL can infer non-time-lagged relationships, referred to as instant causal relationships. This assumes that the current expression level of a gene results from its previous expression level and the current expression level of its causes. These data were generated in a similar manner as the time-lagged except for the method used to generate $X_{i}^{cur}$ and $X_{i}^{sub}$. For instant simulation, we considered Equation (Equation 2), where $causalsub(X_{i})$ is the subsequent value of the causes of $X_{i}$.
(2)$$\begin{array}{c} \begin{array}{cc} X_{i}^{cur} & =f_{1}\left(X_{i}^{pre}\right)+f_{2}\left(causalcur\left(X_{i}\right)\right)+e^{cur}\\ X_{i}^{sub} & =f_{1}\left(X_{i}^{cur}\right)+f_{2}\left(causalsub\left(X_{i}\right)\right)+e^{sub} \end{array} \end{array}$$

To equally benchmark CITL against Scribe, we tested their performance in simulations under both Qiu’s [8] and our set-ups. Qiu’s simulation was based on a manual-curated network of neurogenesis with 12 genes forming 15 directed pairs and two bidirectional pairs. The data were simulated according to the differential equations of these genes at three noise levels. The noise terms followed normal distributions of N(0,0.01), N(0,1), and N(5,0.01) in the low-noise, high-deviation, and high-mean simulation, respectively.

Different gene regulations could have different reaction times in real data sets, resulting in asynchronous changing expression levels of genes. Therefore, we tested the robustness of CITL with the simulations where not all genes were synchronous. For each asynchronous gene *i*, we changed its real changing expression level ($X_{i}^{cha}$) to an asynchronous changing expression level ($X_{i}^{cha-asy}$) by
(3)$$X_{i}^{cha-asy}=a_{i}*X_{i}^{cha},i\in \left\{asynchronous\;genes\right\}$$$a_{i}$ is the random bias value from (1−d) to (1+d). The changing expression levels of asynchronous genes were replaced with $X^{cha-asy}$ in CITL. We tested different proportions of asynchronous genes (0.2, 0.4, 0.6, 0.8, 1.0) and different *d* (0.1, 0.3, 0.5, 0.7, and 0.9) to evaluate the influence of varying degrees of desynchronization on CITL. We carried out 500 single-trace simulations for each set-up. In all simulations, the *k* of the CI test was equal to the square root of the number of genes *n*.

### 4.4. Evaluation

We used precision, recall, and F-measure for the inferred node adjacency versus the data-generating model as the primary evaluation measures to compare the performances of different approaches. To compute these metrics, we first calculated three basic statistics: true positives (TP), false positives (FP), and false negatives (FN) that are related to inferring edges. TP is the number of adjacencies in both the output graph $G_{output}$ from an analytical approach and the true graph $G_{true}$. FP represents the number of adjacencies in $G_{output}$ but not in $G_{true}$. FN is the number of adjacencies in $G_{true}$ but not in $G_{output}$. Precision is the ratio TP/(TP+FP), recall is the ratio TP/(TP + FN), and F-measure is the ratio 2 × precision × recall/(precision + recall). For evaluating the directions, ${\mathrm{TP}}_{direction}$ represented the number of directed edges in both $G_{output}$ and $G_{true}$ with consistent directions. FP represents the number of inconsistent edges in $G_{output}$ compared with $G_{true}$, including absent, undirected, and reverse. FN represents the number of edges in $G_{true}$ but not correctly directed in $G_{output}$. These three metrics have been used in many studies [25,26,27,28,29]. In addition, we used the ability of determining directions (ADD) to evaluate how well a method was able to define directions given true causal edges. ADD was calculated by ${\mathrm{TP}}_{direction}$/${\mathrm{TP}}_{edge}$. Receiver operating characteristic (ROC) [30,31] and precision-recall (PR) [32] curves were also used to evaluate Scribe and Approach 0. However, CITL is unsuitable for ROC or PR curve evaluation because it outputs a binary result rather than a continuous score for each edge.

### 4.5. Real Data Sets

We considered two data sets. Data set 1 was from mouse P0 and P5 dentate gyrus [33], and RNA velocity information was estimated with the same parameters as the example dentate gyrus in Velocyto (http://velocyto.org/ (accessed on 23 May 2018)). There were more than 18,000 cells and an average of 2160 genes for each cell in data set 1 after preprocessing. Data set 2 was the human week ten fetal forebrain data set in Velocyto, containing 1720 cells and an average of 1488 genes for each cell. According to La Manno et al. (2018), the forebrain, as identified by pre-defined markers, can be divided into eight developing stages (0–7). The stage information was only exploited in data visualization.

## 5. Conclusions

In this article, we propose CITL to infer the time-lagged causality of genes using scRNA-seq data. Specifically, we adopted the changing information of genes estimated by RNA velocity in our approach. We further present the superior performance of CITL against other methods in simulations under different set-ups. The proposed approach CITL achieves promising results on a human fetal forebrain scRNA-seq data set, which accurately provides time-lagged causal gene pairs curated by published articles. We note that most methods for analyzing scRNA-seq data did not consider the relationships between genes that could be time-lagged. The results of simulations and real data sets from this paper suggest that we cannot ignore such common relationships. Therefore, we foresee that CITL can provide more insights that may help to guide future gene regulatory research.

## Figures and Tables

**Figure 1 ijms-23-03348-f001:**
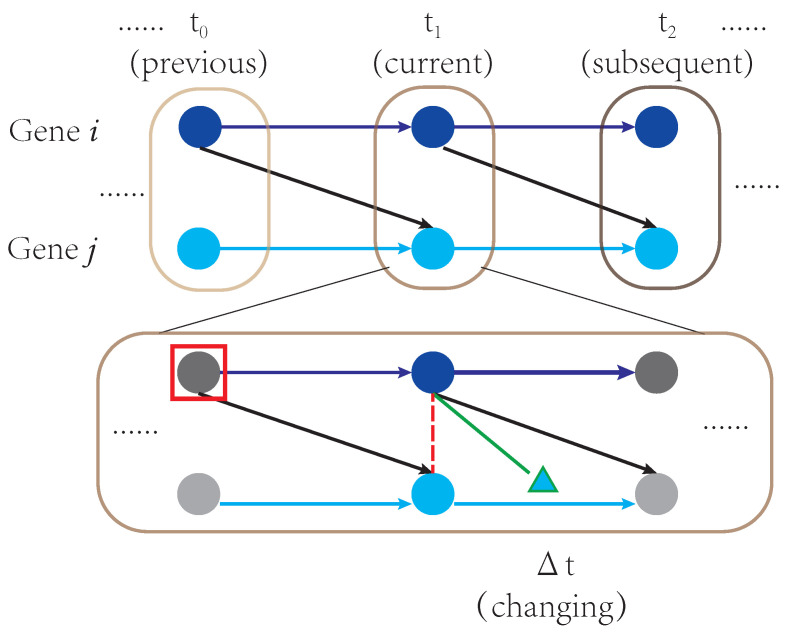
Illustration of a time-lagged relationship across three time points. The gray rectangles represent different individual cells. Multi-trace measurements of three cells (**top**) and one cell’s continuous measurements (**bottom**) are shown. The colored circles represent measurable gene expression levels, while the grey circles represent unmeasurable gene expression levels. The triangle represents the changing expression level of gene *j* in Δt. The blue and black arrows represent true causal relationships between genes across time. The green bar represents the causality between the current and changing expression levels. The red dotted line represents the relationship between the current expression levels of genes, confounded by the previous expression level of gene *i* framed by the red rectangle.

**Figure 2 ijms-23-03348-f002:**
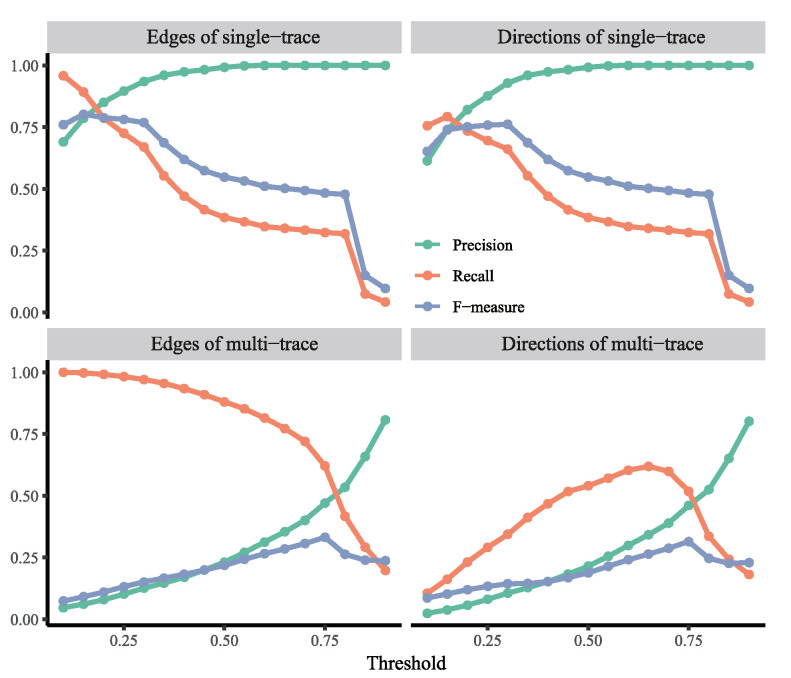
Results of Approach 0 for single-trace simulations (**top row**) and multi-trace simulations (**bottom row**) at different thresholds of “the strong correlation”.

**Figure 3 ijms-23-03348-f003:**
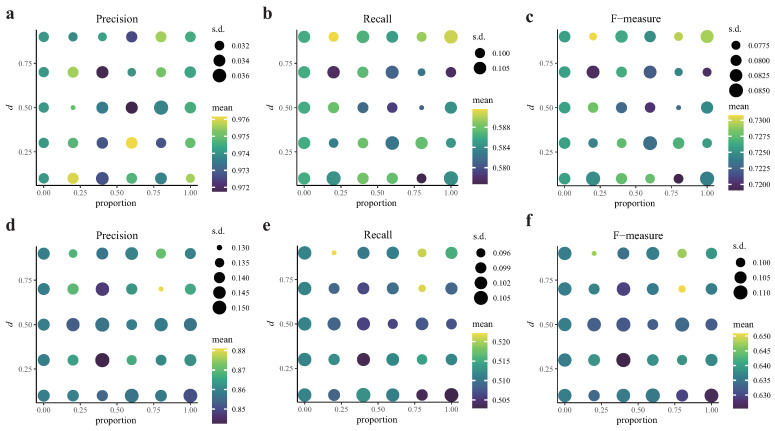
Comparisons of different set-ups of asynchronous simulations. (**a**–**c**) The results of discovering causal pairs. (**d**–**f**) The results of determining causal directions. *d* indicates the range of the random bias.

**Figure 4 ijms-23-03348-f004:**
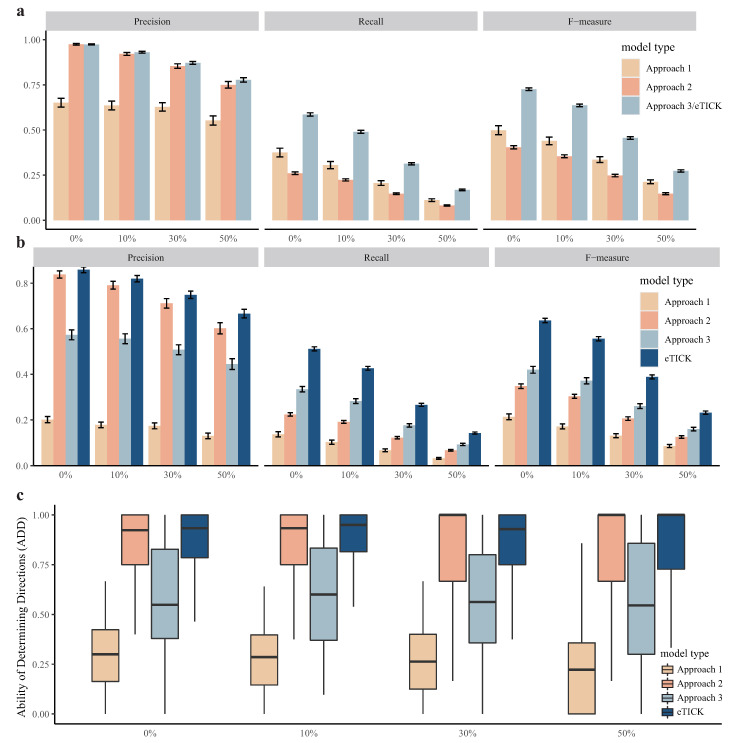
Results of single-trace simulation with latent variables. (**a**) The performance of discovering edges of all approaches with different proportions of latent variables. (**b**) The performance of determining directions of all approaches with different proportions of latent variables. (**c**) The ADD of all approaches with different proportions of latent variables.

**Figure 5 ijms-23-03348-f005:**
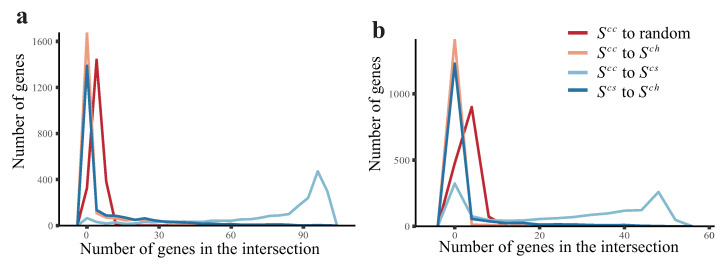
The distribution of intersections as measured by different methods in two data sets. We compared the most correlated genes derived from $X^{cur}$, $X^{cha}$, and $X^{sub}$. Pearson’s correlation coefficient is used to describe three different correlations: (i) correlation between $X_{i}^{cur}$ and $X_{j}^{cur}$ (denoted as cur_cur), (ii) correlation between $X_{i}^{cur}$ and $X_{j}^{sub}$ (denoted as cur_sub), and (iii) correlation between $X_{i}^{cur}$ and $X_{j}^{cha}$ (denoted as cur_cha). For each gene, 100 genes with the largest (50 positive and 50 negative) cur_cur, cur_sub, or cur_cha values were collected into three gene sets $S^{cc}$, $S^{cs}$, and $S^{ch}$, respectively. We recorded the number of genes in the intersections between the three sets. In addition, we recorded the intersection between $S^{cc}$ and 100 randomly selected genes as controls. (**a**) The distribution in data set 1. (**b**) The distribution in data set 2.

**Figure 6 ijms-23-03348-f006:**
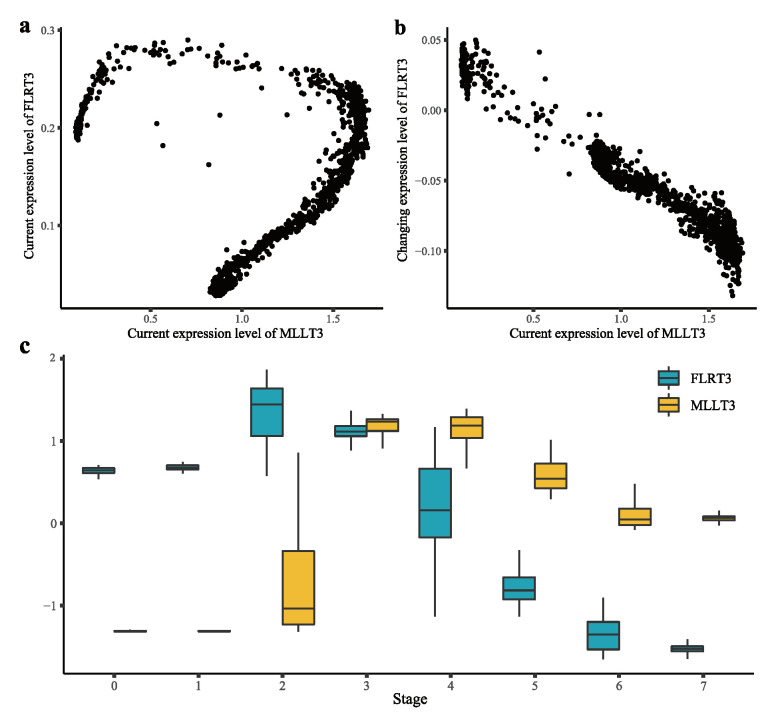
*MLLT3* → *FLRT3*. (**a**) Scatter plot of the current expression levels of *MLLT3* and *FLRT3*. (**b**) Scatter plot of the current expression level of *MLLT3* and the changing expression level of *FLRT3*. (**c**) Box plots of the normalized current expression levels of *MLLT3* and *FLRT3* at eight stages, which was identified by pre-defined markers.

**Table 1 ijms-23-03348-t001:** Comparisons of different approaches based on PC-stable.

**Edges**	**Approach 1**	**Approach 2**	**Approach 3/CITL**	
Precision	0.651 (0.279)	0.975 (0.049)	0.974 (0.033)	
Recall	0.434 (0.275)	0.262 (0.081)	0.597 (0.104)	
F-measure	0.540 (0.264)	0.407 (0.099)	0.734 (0.081)	
**Directions**	**Approach 1**	**Approach 2**	**Approach 3**	**CITL**
Precision	0.201 (0.154)	0.838 (0.180)	0.573 (0.247)	0.859 (0.143)
Recall	0.137 (0.132)	0.224 (0.084)	0.334 (0.137)	0.512 (0.104)
F-measure	0.213 (0.124)	0.348 (0.113)	0.420 (0.169)	0.636 (0.108)

The average values from 500 single-trace simulations are shown with standard deviation values in parentheses.

**Table 2 ijms-23-03348-t002:** Comparisons between CITL and Scribe in single-trace and multi-trace simulations.

**Single-Trace**	**CITL**	**Scribe (Top)**	**Scribe (AUROC)**	**Scribe (AUPR)**
**Discovered edges**	21	21	0.984	0.864
**Real edges**	21	21		
**Correct directions**	21	21		
**Multi-trace**	**CITL**	**Scribe (top)**	**Scribe (AUROC)**	**Scribe (AUPR)**
**Discovered edges**	22	22	0.820	0.534
**Real edges**	22	22		
**Correct directions**	22	18		

sd: standard deviation.

**Table 3 ijms-23-03348-t003:** Detailed paths of the causal pairs with regulatory relationships.

rTLP	Path	cur_cur	cur_cha	cur_sub	Type of Each Step
*TCF7L2* → *FTH1*	*TCF7L2* → *RELA* → *FTH1*	−0.98	0.96	−0.98	PROT, TRSC
*YBX1* → *EPS8*	*YBX1* → NF-κ-B signaling → *EPS8*	0.04	0.95	0.54	PROT, TRSC
*NFIA* ↛ *EIF4A2*	*NFIA* ↛ ERK pathways ↛ *PPARGC1A* ↛ *EIF4A2*	0.81	−0.98	0.81	PROT, TRSC, TRSC
*SATB2* ↛ *TUBB2B*	*SATB2* ↛ MAPK7 → NEUROG2 → *TUBB2B*	0.71	−0.95	0.71	PROT, PROT, TRSC
*VHL* ↛ *ABCD2*	*VHL* ↛ WNT → AMPK → *ABCD2*	0.63	−0.58	0.51	PROT, PROT or TRSC, TRSC
*VHL* ↛ *PCDH9*	*VHL* → *TP53* ↛ *PCDH9*	0.39	−0.88	−0.26	TRSC, TRSC
*TSC22D1* ↛ *PTPRD*	*TSC22D1* ↛ MTOR ↛ MYCN ↛ *PTPRD*	0.03	−0.92	−0.94	PROT, PROT, TRSC
*TSC22D1* ↛ *ZBTB18*	*TSC22D1* → TGFB1 ↛ ASCL1 → *ZBTB18*	0.01	−0.84	−0.0031	PROT or TRSC, PROT, TRSC
*MLLT3* ↛ *FLRT3*	*MLLT3* → E2F1 ↛ WNT → *FLRT3*	−0.19	−0.99	−0.65	PROT, PROT, TRSC
*NFKB1* → *HSPA8*	*NFKB1* → *MYB* → *HSPA8*	0.69	0.99	0.95	TRSC, TRSC
*SFPQ* ↛ *JUND*	*SFPQ* ↛ PGR → *JUND*	−0.67	−0.96	−0.69	PROT, TRSC
*HDAC2* ↛ *HSP90AA1*	*HDAC2* ↛ *MIR15A* ↛ *HSP90AA1*	−0.80	−1.00	−0.84	TRSC, TRSC
*TSC22D1* ↛ *KDM5B*	*TSC22D1* → transforming growth factor → *KDM5B*	0.40	−0.82	0.35	PROT, TRSC
*EOMES* → *MAGED1*	*EOMES* ← ERK1/2 ← *MAGED1*	−0.03	0.97	0.29	TRSC, PROT

cur_cur: the Pearson’s correlation coefficient between the current expression levels of the cause and the target; cur_cha: the Pearson’s correlation coefficient between the current expression level of the cause and the changing expression level of the target; cur_sub: the Pearson’s correlation coefficient between the current expression level of the cause and the subsequent expression level of the target. →: up-regulation. ↛: down-regulation.

## Data Availability

Publicly available datasets were analyzed in this study. This data can be found here: http://velocyto.org/ (accessed on 10 January 2022).

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
