# Peer review of "Inferring Time-Lagged Causality Using the Derivative of Single-Cell Expression"

_ijms, 2022, doi:10.3390/ijms23063348_

Round 1

Reviewer 1 Report

Developing novel bioinformatics tools that allow expression level analysis of RNA, especially at the single-cell resolution, and infer causality among genes is crucial. And this vital topic was taken up by Wei et al.  They developed the CITL bioinformatics tool that can infer time-lagged causal relationships from scRNA-seq data by assessing conditional independence between the changing and current expression levels of genes. It is justified taking into account the number of rapidly growing scRNA-seq based studies in the databases. The reviewed article is very well written and may be published in IJMS. Before publication, however, a few changes to the manuscript should be made. 

Line 186-192 is the repetition of lines 181-186.

Line 223 and 224: Repetition of “Results”. Consequently, the numbering of the Results subsections should also be corrected.
Line 266: Repeated “Table A”.

In supplementary data, please indicate if there is any specific meaning of numbers in red font.

Line 186-192 is the repetition of lines 181-186.

Line 223 and 224: Repetition of “Results”. Consequently, the numbering of the Results subsections should also be corrected.

Line 266: Repeated “Table A”.

In supplementary data, please indicate if there is any specific meaning of numbers in red font.

A few things need to be added on the tools’ page or readme file: An average user may not understand "R" on the Github page. You can add an explanation, a fuller name for this "R", and a message that it should be downloaded and installed separately.

The CITL page provides instructions for command tools for CITL and R packages. It is confusing whether both are required for efficiently running the tool. Any difference between these two should be specified.

Also, I tried running both the command line as well as the R version of the tool. The instructions for running the R package worked fine but could not get command tools to work. On executing the following command from the analysis directory (on a Linux OS):
Rscript CITL.R ../test/Spliced_test.csv ../test/delta_s_test.csv ../test/gene_names_test.csv 15

A "Connection time out" message was displayed. Authors should check if any additional information is required for command tools to work properly.

Author Response

Response to Reviewer 1

Thank you for your suggestions about correcting the repetition.

Comment 1

Line 186-192 is the repetition of lines 181-186.

Resopnse 1

Line 186-192 had been deleted.

Comment 2

Line 223 and 224: Repetition of “Results”. Consequently, the numbering of the Results subsections should also be corrected.

Resopnse 2

The repeated “Results” had been deleted. Further, the numbering of the following sections has been revised.

Comment 3

Line 266: Repeated “Table A”.

Response 3

Line 226: The repeated “Table A” had been deleted.

Comment 4

In supplementary data, please indicate if there is any specific meaning of numbers in red font.

Response 4

The numbers in red font in supplementary tables indicate significantly larger than others (t-test). According to your advice, we had supplemented it in footnotes.

Comment 5

A few things need to be added on the tools’ page or readme file: An average user may not understand "R" on the Github page. You can add an explanation, a fuller name for this "R", and a message that it should be downloaded and installed separately.

Response 5

We are appreciated for your suggestions and error reports about the CITL tools.

We have included an introduction to R and its installation guide in the readme files of the command-line tool and the R package.

Comment 6

The CITL page provides instructions for command tools for CITL and R packages. It is confusing whether both are required for efficiently running the tool. Any difference between these two should be specified.

Response 6

Both tools can run independently. We describe the recommended scenario for each tool in the readme file.

Comment 7

Also, I tried running both the command line as well as the R version of the tool. The instructions for running the R package worked fine but could not get command tools to work. On executing the following command from the analysis directory (on a Linux OS):

Rscript CITL.R ../test/Spliced_test.csv ../test/delta_s_test.csv ../test/gene_names_test.csv 15

A "Connection time out" message was displayed. Authors should check if any additional information is required for command tools to work properly.

Response 7

The error “Connection time out” has been fixed.

Reviewer 2 Report

In this study, the authors propose an infer time-lagged causality among genes using cross-sectional single-cell RNA sequencing (scRNA-seq) data. The study is reasonable and the authors released the package/tool for their method. It holds potential for publication, but some major points should be addressed:

1. Besides current metrics, the authors should show the ROC & PR curves of the models.

2. The authors should compare the performance results to previously published works on the same problem/data.

3. More discussions should be added.

4. Which statistical tests are used in Fig. 4 (as shown in error bars)?

5. Measurement metrics (i.e., recall, precision, ...) have been used in previous bioinformatics studies such as PMID: 34502160, PMID: 34915158. Therefore, the authors are suggested to refer to more works in this description to attract a broader readership.

6. Line 223-224: there are 2 sections "Results".

7. Why did the first approach number as '0' rather than '1'?

8. English language and presentation style should be minor checked.

Author Response

Response to Reviewer 2

Comment 1

  1. Besides current metrics, the authors should show the ROC & PR curves of the models.

Response 1

Thank you for the suggestions about including the models’ receiver operating characteristic (ROC) & precision-recall (PR) curves. It should be noticed that the number of non-existing edges (true negatives) outweighs the number of existing edges (true positives) in causal networks significantly. Therefore, the PR curves are more informative [1].

“Scribe” and “Approach 0” calculate some kinds of scores for each edge to measure its probability of being a causal edge in the networks. These models are suitable for evaluation by ROC & PR curves. We have included the ROC & PR curves of the two models in the manuscript and the supplementary notes. The revised results are different from the previous because we found and fixed some bugs when testing Scribe.

However, the conditional independence tests determine edges in the networks in other approaches, including Approach 1, 2, 3, and CITL. Under a certain significance level, the outputs of these approaches are whether an edge exists or not. The binary results are not suitable with ROC or PR curves. We tried to use different significance levels, although the significance level should not be tuned like the threshold in ROC & PR curves. The performance of the approaches does not vary very much (Table 1).       

Alpha

Number of true edges

Discovered edges

Real edges

Correct directions

0.8

53

26

26

23

0.504766

53

30

29

27

0.318486

53

28

28

27

0.200951

53

28

28

28

0.126791

53

27

27

27

0.08

53

26

26

26

0.050477

53

25

25

25

0.031849

53

25

25

25

0.020095

53

25

25

25

0.012679

53

25

25

25

0.008

53

25

25

25

0.005048

53

25

25

25

0.003185

53

25

25

25

0.00201

53

24

24

24

0.001268

53

24

24

24

8.00E-04

53

23

23

23

0.000505

53

22

22

22

0.000318

53

23

23

23

0.000201

53

23

23

23

0.000127

53

23

23

23

8.00E-05

53

23

23

23

5.05E-05

53

23

23

23

3.18E-05

53

23

23

23

2.01E-05

53

23

23

23

1.27E-05

53

23

23

23

8.00E-06

53

23

23

23

5.05E-06

53

23

23

23

3.18E-06

53

22

22

22

2.01E-06

53

22

22

22

1.27E-06

53

22

22

22

8.00E-07

53

22

22

22

5.05E-07

53

22

22

22

3.18E-07

53

22

22

22

2.01E-07

53

22

22

22

1.27E-07

53

22

22

22

8.00E-08

53

22

22

22

5.05E-08

53

22

22

22

3.18E-08

53

22

22

22

2.01E-08

53

21

21

21

1.27E-08

53

21

21

21

8.00E-09

53

21

21

21

5.05E-09

53

21

21

21

3.18E-09

53

21

21

21

2.01E-09

53

21

21

21

1.27E-09

53

21

21

21

8.00E-10

53

21

21

21

Table 1 The performance of CITL at different significance levels.

Comment 2

  1. The authors should compare the performance results to previously published works on the same problem/data.

Response 2

In the simulations, the comparison between the previously published work (Scribe) and CITL under different settings is described in the revised Section 3.1.3. On the real data sets, we did not use Scribe because it ran too slow for a 1400-node graph.

Comment 3

  1. More discussions should be added.

Response 3

We have included more discussion about the difference between Scribe and CITL.

Comment 4

  1. Which statistical tests are used in Fig. 4 (as shown in error bars)?

Response 4

Error bars in Fig. 4 represent the standard deviation of the corresponding values.

Comment 5

  1. Measurement metrics (i.e., recall, precision, ...) have been used in previous bioinformatics studies such as PMID: 34502160, PMID: 34915158. Therefore, the authors are suggested to refer to more works in this description to attract a broader readership.

Response 5

Thank you for advising us to include some related works. We have referred to more works in the manuscript.

Comment 6

  1. Line 223-224: there are 2 sections "Results".

Response 6

The replicated words have been deleted.

Comment 7

  1. Why did the first approach number as '0' rather than '1'?

Response 7

Approach 0 is not a rigorous approach but a straightforward implementation of the Time-lagged Assumption. It is to assist in understanding the information in the changing expression of genes. Therefore, it is numbered as 0.

Comment 8

  1. English language and presentation style should be minor checked.

Response 8

We have revised some language and presentation styles.

[1] Saito T, Rehmsmeier M. The precision-recall plot is more informative than the ROC plot when evaluating binary classifiers on imbalanced datasets. PLoS One. 2015 Mar 4;10(3):e0118432.

Round 2

Reviewer 2 Report

My previous comments have been addressed well.